# The Impact of Cyclic Oxidation in Dissociated Air on the Mechanical Properties of Freeze-Cast ZrB_2_/MoSi_2_ Ceramics

**DOI:** 10.3390/ma18081815

**Published:** 2025-04-15

**Authors:** Ludovic Charpentier, Eric Bêche, Hervé Glénat, Álvaro Sández-Gómez, Pedro Miranda

**Affiliations:** 1PROMES, CNRS, 7, Rue Du Four Solaire, 66120 Font-Romeu Odeillo, France; eric.beche@promes.cnrs.fr (E.B.); herve.glenat@promes.cnrs.fr (H.G.); 2Department of Mechanical, Energy and Materials Engineering, Industrial Engineering School, University of Extremadura, Avda. de Elvas s/n, 06006 Badajoz, Spain; alvarosg@unex.es (Á.S.-G.); pmiranda@unex.es (P.M.)

**Keywords:** oxidation, dissociated atmosphere, scanning electron microscopy, nanoindentation

## Abstract

Creating reusable thermal shields would decrease our carbon footprint by eliminating the need for the reapplication of single-use ablative alternatives. Our previous investigations identified ultra-high-temperature ZrB_2_ with 20 vol.% MoSi_2_ ceramics as a promising candidate for the fabrication of reusable thermal shields. Therefore, in this study, this material was exposed to cyclic oxidation at 1800 and 2150 K in dissociated air in order to investigate how it might withstand multiple terrestrial re-entries. At 1800 K, we observed semi-parabolic oxidation kinetics with the growth of a protective oxide layer, the silica-based composition of which was determined using XRD and SEM (coupled with EDS). More dramatic damage was observed at 2150 K, with continuous linear oxidation kinetics seen. Cross-section hardness measurements using nanoindentation revealed that the oxidized part of the samples was the source of their mechanical weakness, suggesting that the material should be used below 1800 K to ensure its reusability.

## 1. Introduction

Many industries, such as aerospace or energy generation, would benefit from thermal protection systems that can endure repeated use in aggressive environments. Ultra-high-temperature ceramics such as ZrB_2_ with added Si have been identified in the literature as good candidates for such systems as they can resist oxidation and flexural stresses beyond 2000 K [1,2]. In previous work [3], we identified that ultra-high-temperature ceramics composed of ZrB_2_ with a 20 vol.% MoSi_2_ (hereafter referred to as ZBM) exposed to dissociated air would form a protective layer of silica up to a temperature of 2000 K. We hereby continue this investigation by analyzing the damage that is induced on freeze-cast ZBM samples by several cycles of oxidation below (at 1800 K) and above (at 2150 K) this temperature limit. At 1800 K, the formation of silica is expected to favor a diffusion-controlled oxidation regime, reducing the oxidation rate, as the silica layer acts as a protective barrier while growing [4]. At 2150 K, the absence of silica should favor a reaction-controlled oxidation regime with a constant oxidation rate. This investigation aims to confirm that there is a maximal working temperature below which ZBM could be reused. As an example of such applications, we are developing in the AM-ACTS (Additive Manufacturing—Actively Cooled Thermal Shield) M-era.Net project [5] a ceramic–alloy thermal shield whose working temperature can be adjusted by circulating a coolant through an internal micro-channeled network, ensuring the critical temperature is not exceeded and thus enabling the survival of the thermal shield over multiple cycles of operation with very limited damage.

## 2. Materials and Methods

High-purity commercial powders of grade B ZrB_2_ (94–96 wt.% purity, Hf 2 wt.%, O 1.1 wt.%, C 0.2 wt.%, N 0.05 wt.%, d50 ~1.5–3 µm, ρ = 6.10 g cm^−3^) (Höganäs AB, Höganäs, Sweden) and grade B MoSi_2_ (>99%, d50 ~3.5–4 µm, ρ = 5.9 g cm^−3^) (Höganäs AB, Höganäs, Sweden) were used to prepare the ZrB_2_ + 20% vol. MoSi_2_ mixtures. These were ball-milled in ethanol for 24 h and subsequently dried on a hot plate with continuous magnetic stirring. SEM pictures of the powders before and after ball-milling are presented in Figure 1. The ZBM samples were prepared using a freeze casting process and subsequent pressureless spark plasma sintering (SPS), following a procedure described in detail in a previous study [6]. Basically, aqueous suspensions of the pre-mixed powders, with a 45 vol.% solid content and 1.5 wt.% of polyethyleneimine (PEI delivered by Sigma Aldrich, Merck KGaA, Darmstadt, Germany, 10,000 MW), used as a deflocculant, were poured into a mold and quickly and unidirectionally frozen using a cold metallic rod immersed in liquid nitrogen. Samples were freeze-dried (Lyoquest-55, Telstar Sintegon, Terrassa, Spain) at 50 °C and 0.030 mbar over 24 h and subsequently densified in an SPS furnace (HP-D-10, FCT Systeme GmbH, Frankenblick, Germany) at 1900 °C for 15 min, using a 100 °C min^−1^ ramp and a 30 min plateau at 1500 °C to facilitate the release of boria. The heat treatment was performed without applying pressure to the samples because of the use of appropriate graphite dies. Figure 2 presents the X-ray diffraction patterns of the as-produced sample and the initial powders. It can be observed that the samples present crystalline phases of ZrB_2_ (ICDD-JCPDS card N°75-0964) and MoSi_2_ (2 ICDD-JCPDS card N°81-2167), similar to those in the initial powders. Some minor MoC and ZrC intergranular inclusions were also identified in previous work [6] on the elaboration and the structural characterization of ZBM samples using SEM and EDX (JEOL Ltd., Tokyo, Japan).

The experimental MESOX (*Moyen d’Essai Solaire d’OXydation*—Oxidation Solar Test Facility) in the PROMES-CNRS laboratory [7,8,9,10] was used to oxidize the freeze-cast ZBM samples at a total atmospheric pressure of 1000 Pa and an air flow of 4 L h^−1^. A microwave generator was used to dissociate the oxygen molecules in the MESOX. The output power of the generator was set to 300 W so that 70% of the oxygen molecules could be dissociated [7]. Cyclic oxidations were performed at 1800 and 2150 K. A single ZBM sample was held at each temperature for 5 min and then cooled down, weighted, and repositioned for the following cycle. The plateau duration was chosen in order to correspond to the duration when a thermal shield is exposed to dissociated oxygen during a terrestrial re-entry and to be long enough to observe significant changes due to the oxidation [3]. Up to 10 oxidation cycles were performed at 1800 K, and up to 5 at 2150 K. This number of cycles was chosen to simulate the impact of several re-entries on a thermal shield. The samples were weighed before and after each oxidation cycle, and their variation in mass due to surface oxidation was reported in mg cm^−2^. The accuracy of the temperature and weight measurements was ±20 K and ±0.1 mg, respectively.

X-ray diffraction (XRD) analyses were performed at room temperature on the oxidized surfaces using a PANalytical (Malvern Panalytical Ltd, Malvern, UK) diffractometer XPert Pro (CuKα radiation, λ = 0.15418 nm). XRD measurements of θ-θ symmetrical scans were made over an angular range of 10 to 80°. The step size and the time per step were fixed at 0.017° and 70 s, respectively. The X-ray diffractograms were recorded and studied using PANalytical software (Datacollector and HighScorePlus). The contribution of the AlKα2 diffractograms was removed using the Rachinger method [11]. The instrumental function was determined using a reference material (SRM 660, lanthanum hexaboride, LaB6 polycrystalline sample) and can be expressed as a polynomial function [12]. Scanning electron microscopy coupled with energy-dispersive X-ray spectroscopy (SEM/EDS) was carried out on the surfaces and cross-sections of the oxidized samples using JEOL IT800 SHL LV equipment. Cross-section images of the samples were taken after cutting the oxidized specimens along their diameter, embedding them in resin, and polishing the cut surface down to a 1 µm finish.

The mechanical properties were measured on sections of the as-received and oxidized ZBM samples with a G200 X model KLA nanoindenter (KLA Corporation, Milpitas, CA, USA) using a diamond Berkovich tip. The samples were analyzed following the NanoBlitz 3D method developed by the KLA Corporation. This technique, which relies on fast indentation tests, can spatially map the mechanical properties of a tested material by performing up to 40,000 indentations. A specific load is applied to the sample’s surface, and the resulting depth of the indentation and contact stiffness are acquired and then converted into the hardness and elastic modulus of the material using the Oliver–Pharr model [13,14]. The mechanical mappings presented here were achieved with an applied load of 3.5 mN. The elastic modulus was calculated with a Poisson’s ratio corresponding to that of ZrB2, i.e., 0.14. The indentation network was constructed using the following parameters: 20 indentations over 100 µm along the X axis, 80 indentations over 400 µm along the Y axis. These parameters were chosen in order to ensure both a good resolution for the mapping (the closer the indentations are, the better the resolution) and to avoid any disturbances due to the plasticity induced by the indentations (which requires sufficient spacing between the indentations). These conditions can be satisfied if the spacing between two adjacent indents is greater than three times the contact diameter. The sections of the ZBM samples were also observed using a numerical microscope: Keyence VHX-7000 (Keyence Corporation, Osaka, Japan).

## 3. Results

### 3.1. Oxidation Kinetics

Figure 3a presents the evolution of the weight changes per unit surface (in mg cm^−2^) with time at 1800 and 2150 K. The data at the highest temperature can be fitted with a linear regression, which is not the case for the data at 1800 K, which appear to follow a square root trend. Therefore, in Figure 3b we have mapped the evolution of the square of the weight changes at 1800 K, which follows a linear trend. Thus, Figure 3 proves that the oxidation kinetic is semi-parabolic at 1800 K (Equation (1)) and linear at 2150 K (Equation (2)):Δm/S = k_p_·t^1/2^ where k_p_ = (5.16)^1/2^ = 2.27 mg cm^−2^ min^−0.5^ at 1800 K (1)Δm/S = k_l_·t where k_l_ = 0.89 mg cm^−2^ min^−1^ at 2150 K (2)

The oxidation mechanism of a compound is mainly controlled by two phenomena: the diffusion of oxidizing species and gaseous products through the continuously growing oxide layer and the rate of the oxide’s production. According to the general law established by Deal and Grove [2], the oxidation kinetics of a compound are linear when controlled by the reaction rate and semi-parabolic when controlled by diffusion. Thus, the oxidation of the ZBM samples in dissociated air at 1000 Pa is diffusion-controlled at 1800 K but reaction-controlled at 2150 K.

### 3.2. Post-Experimental Characterizations

#### 3.2.1. Oxidation at 1800 K

The ZBM samples oxidized at 1800 K were characterized using XRD after one cycle (5 min), five cycles (25 min), and ten cycles (50 min) of oxidation and compared with an as-received sample. Figure 4a presents the complete diffractograms of each sample and Figure 4b shows a close-up comparison of the 16–26° area at the three oxidation times.

The ZrB_2_ and MoSi_2_ phases were not detected on the surface of the samples after one oxidation cycle, proving that a complete oxide layer had formed with monoclinic zirconia (ICDD-JCPDS card N°78-0048 or 70-8739) as its main crystalline phase. Tetragonal zirconia (ICDD-JCPDS card N°72-2743) and tetragonal molybdenum boride (ICDD-JCPDS card N°51-0940 or 73-1768) were detected as minor crystalline phases. The presence of an amorphous oxide is also evidenced by the bump observed in Figure 4b, which increases with the oxidation time.

The surface SEM images and associated EDS analyses of two different spots are shown in Figure 5. The as-received sample has a rough surface (Figure 5a) made up of ZrB_2_ grains (darker areas, EDS in Figure 5b) and a MoSi_2_ intergranular phase (clear areas, EDS in Figure 5c). All of the oxidized samples (Figure 5d,g,j) display a full-cover vitreous phase with some clearer grains encapsulated within it. The EDS shows that both the amorphous and crystalline phases are oxides. Figure 5e,h,k show that the vitreous phase mainly contains Si and O elements in a proportion close to that of SiO_2_. Figure 5f,i,k demonstrate that the encapsulated grains are richer in zirconium. By comparing these results with those from the XRD, we can conclude that the vitreous phase is the amorphous phase displayed in Figure 4b and that the encapsulated grains correspond to the identified monoclinic and (minor) tetragonal ZrO_2_ phases. Crucially, we can also observe the presence of molybdenum in Figure 5l, which corresponds to the MoB phases identified through XRD. Some gaseous molybdenum trioxide may have also condensed on the sample during the cooling phase.

Finally, Figure 6 presents SEM images of the cross-section of the oxidized ZBM samples and the associated EDS mapping of their Si, O, Zr, and B elements. All of the samples contain three visibly different areas:An unaltered substrate (with the presence of boron and absence of oxygen) in the deeper part of the sample;An intermediate oxide layer (as evidenced by the presence of oxygen) covering the substrate, which is characterized by a reduced amount of silicon and boron. This phase grows with the oxidation time: its thickness was 25, 30, and 65 µm after one, five, and ten oxidation cycles, respectively;A thin outer region rich in silicon and oxygen at the surface of the samples. We observed that this amorphous phase forms as a few separated droplets after one oxidation cycle, a discontinuous thin layer after five oxidation cycles, and a 20 µm-thick continuous outer crust after ten oxidation cycles.

**Figure 6 materials-18-01815-f006:**
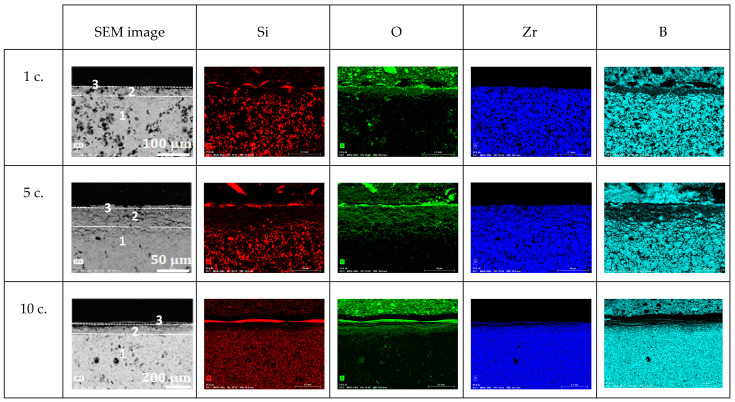
Cross-section SEM images (BSE mode) and EDS mapping of Si, O, Zr, and B elements in ZBM samples after 1, 5, and 10 oxidation cycles at 1800 K in dissociated air at P = 1000 Pa.

#### 3.2.2. Oxidation at 2150 K

Figure 7 presents the surface SEM images and EDS analyses of the ZBM samples after one or five oxidation cycles at 2150 K (the as-received surface was similar to the one presented in Figure 5a–c and is not shown for brevity). After one oxidation cycle (Figure 7a,b), the sample’s irregular surface is covered in crystalline grains and is abundantly microporous. The EDS shows that zirconium, boron, oxygen, and carbon are the predominant species at the surface of the oxidized sample. After five oxidation cycles (Figure 7c–f), the surface looks more heterogeneous and can be interpreted as follows. Firstly, the clearer grains (Figure 7d) mainly consist of zirconium and oxygen. Carbon and silicon are also present in significant amounts. Secondly, the darkest areas (Figure 7e) contain significant amounts of silicon, oxygen, and boron. Zirconium is nearly absent in these areas. Thirdly, the dark gray regions (Figure 7f) contain significant amounts of silicon, oxygen, and zirconium. No boron is detected in these areas.

Figure 8 displays the XRD of the ZBM samples after five oxidation cycles at 2150 K. The main crystalline phase that was identified is monoclinic zirconia. Tetragonal zirconia is present in small amounts. Zooming in on the 16–27° 2θ range (Figure 8b) reveals a slight bump corresponding to the presence of an amorphous phase.

Figure 9 presents the cross-section SEM images of the ZBM samples after one and five oxidation cycles, and the EDS mapping of their Si, O, Zr, and B elements. We can identify two main areas in these samples:The substrate (containing Zr, B, and Si) at the bottom of the SEM images.An oxide layer that only contains O and Zr. After one oxidation cycle, this layer is already ap. 65 µm thick. After five oxidation cycles, this thickness has increased to ap. 275 µm. There is also a fissure crossing the sample, parallel to the external surface and the interface with the substrate, in this layer, which probably appeared during sample cutting and/or polishing. After five oxidation cycles, there is a thin Si-rich layer peeling off the sample with B-rich grains on its external surface.

**Figure 9 materials-18-01815-f009:**
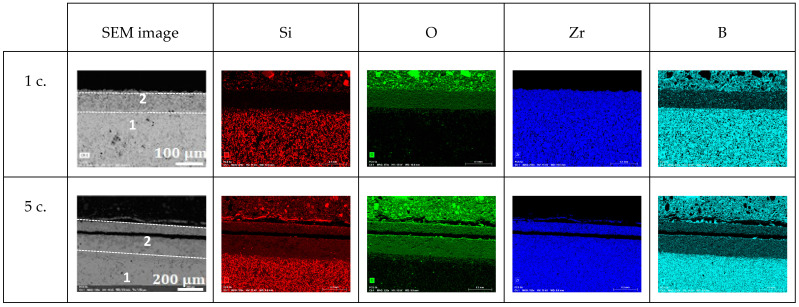
Cross-section SEM images (BSE mode) and EDS mapping of Si, O, Zr, and B elements in ZBM samples after 1 and 5 oxidation cycles at 2150 K in dissociated air at P = 1000 Pa.

### 3.3. Mechanical Properties

Figure 10 shows the maps of the elastic modulus (E) and hardness (H) measured across the selected section of the as-received ZBM sample. In Figure 10, Figure 11 and Figure 12, the Y position at 400 µm corresponds to the top of the samples’ surface. As observed in Figure 10b, the modulus and hardness of the sample are homogeneous along its depth (Y axis), with averages of 471 ± 89 GPa and 17 ± 7.6 GPa, respectively. The significant standard deviation of the measurements is mainly related to the local chemical inhomogeneity of the ZBM. Other effects are also likely to contribute to this deviation, such as the crystallographic orientation of the powders and the roughness of the analyzed surfaces. Indeed, despite careful polishing of the samples, the presence of microporosities inside the ZBM causes holes to appear on the polished surfaces, particularly in the sample oxidized at 2150 K, which can lead to inaccuracies in determining the contact surface between the Berkovich indenter and the sample [15]. Nevertheless, the very large number of indentations per map provides satisfactory statistical measurements that are in agreement with those published previously. Pure ZrB_2_ is a very stiff material with a modulus of elasticity reported to be around 500 GPa [16]. Using a mechanical characterization method that made use of the impulse excitation of resonant vibrations, R. J. Grohsmeyer et al. showed that the modulus of ZrB_2_ progressively decreases with an increasing MoSi_2_ content [17].

Figure 11 shows similar maps of sections of the ZBM sample subjected to 10 oxidation cycles at 1800 K in dissociated air at P = 1000 Pa. The evolution of its elastic modulus and hardness along the Y axis, presented in Figure 11b, highlights the presence of a layer with low resistance to deformation (E = 56 ± 6.6 GPa; H = 0.95 ± 0.28) of about a 50 µm thickness, followed by a transition area of about a 50 µm thickness which progressively takes on the mechanical properties of the harder bulk ZBM (E = 462 ± 86 GPa; H = 16.7 ± 6.8 GPa). As with the SEM cross-section images in Figure 6, we can identify a clear separation between the hard substrate and the oxide layer.

Finally, the maps of the elastic modulus and hardness throughout the section of the ZBM sample subjected to five oxidation cycles at 2150 K are shown in Figure 12. The area with a low resistance to deformation (E = 64 ± 21 GPa; H = 1.47 ± 0.76 GPa) in Figure 12b is thicker than that in Figure 11b and reaches a thickness of about 200 µm. It is always followed by a transition area of about a 50 µm thickness that gradually takes on the mechanical properties of the harder bulk ZBM (E = 491 ± 89 GPa; H = 20 ± 6.6 GPa). 

## 4. Discussion

The characterizations performed provide results that agree with our own previous observations [3] and those from other authors [18,19]: ZrB_2_ oxidizes into zirconia at any temperature in accordance with Equation (3), and MoSi_2_ oxidizes into either SiO_2_ (at 1800 K) or gaseous SiO (at 2150 K) in accordance with Equation (4) or (5).ZrB_2_(s) + 5 O^●^ → ZrO_2_(s) + B_2_O_3_(l/g)(3)MoSi_2_ + 7 O^●^ → 2SiO_2_(s) + MoO_3_(g)(4)MoSi_2_ + 5 O^●^ → 2SiO(g) + MoO_3_(g)(5)

The surface and cross-section SEM images of the ZBM samples oxidized at 1800 K and their corresponding EDS maps show that the silica produced according to Equation (4) migrates to the top of the oxide layer that becomes thicker and more continuous with time. After 10 oxidation cycles, corresponding to a 50 min total exposure to high temperatures and dissociated oxygen, this surface is 20 µm thick and significantly reduces the rate at which oxygen penetrates the sample. This is the reason why the oxidation is diffusion-controlled with a semi-parabolic oxidation kinetic at 1800 K. This silica layer is likely the amorphous compound whose presence in the XRD results becomes more and more obvious with time. This silica layer covers a zirconia-rich area (whose thickness reaches 65 µm after 10 oxidation cycles) which is, according to the XRD results, mainly monoclinic with some minor tetragonal inclusions. Inclusions of zirconia were also present in the upper glassy silica layer, as shown by the surface SEM and EDS analyses. The surface EDS also identified the presence of molybdenum (which can result from MoO_3_ condensation during cooling), sodium and calcium (probably due to surface contaminations), and carbon and hafnium (due to impurities in the powders and contamination during sintering in SPS, these elements were already present in the as-received sample).

At 2150 K, as silicon oxidizes into a gaseous compound, zirconia alone should cover the substrate. Nevertheless, we observed, using surface and cross-section EDS analyses, some silicon and boron oxide phases in this layer after five oxidation cycles, which correspond to a 25 min total exposure time. As the covering layer is very thin, and considering that 2150 K is far above the known stability domain for both compounds at a total pressure of 1000 Pa [3], the condensation of gaseous oxides during cooling seems the only pertinent explanation for their presence. The microporosity in the zirconia layer (which is mainly monoclinic) favors the diffusion of oxygen and, therefore, the oxidation is reaction-controlled with a linear oxidation kinetic at 2150 K. The zirconia layer can, consequently, become really thick, growing up to 275 µm thick after five oxidation cycles.

The nanoindentation tests performed clearly differentiated the oxidized areas from the substrate. After 10 oxidation cycles at 1800 K, we observed an external layer that was about 50 µm thick and had low resistance to deformation. This layer is supported by a transition area with a thickness of about 50 µm which gradually converges towards the mechanical properties of the bulk ZBM. These two layers roughly correspond to the complete oxide thickness observed during the cross-section SEM and EDS mapping (20 µm of silica + 65 µm of zirconia, i.e., 85 µm in all). After five oxidation cycles at 2150 K, the layer with a low resistance to deformation was 200 µm thick and the transition layer was 50 µm thick, which corresponds to the thickness measured using the SEM coupled with EDS (275 µm). The oxide layer formed at 1800 K is not only more stable but also harder and stiffer than that formed at 2150 K. This can be attributed to the presence of glassy silica at 1800 K, which significantly reduces the porosity of the oxide layer, thus strengthening it. A recent investigation also indicated that the presence of silica could stabilize the formation of harder monoclinic phases [20]. The significant standard deviation in the hardness measurements in the oxide layer can be explained by the inhomogeneity of the oxide, as it includes silica, monoclinic, and tetragonal zirconia at 1800 K and zirconia and porosities at 2150 K. As expected, the substrates’ mechanical properties after thermal treatment at 1800 or 2150 K are identical to those of the as-received ZBM, so there was no significant influence of the temperature on the mechanical properties of the ZBM itself.

## 5. Conclusions

The main outcomes of this investigation are as follows:

(1) At 1800 K, in air at P = 1000 Pa and in the presence of dissociated oxygen, ZrB_2_ + 20 v.% MoSi_2_ oxidizes to form an intermediate oxide layer made of monoclinic zirconia that is covered by a silica layer, reducing the oxidation rate of the substrate by controlling the diffusion of oxygen. This diffusion-controlled mechanism is characterized by semi-parabolic oxidation kinetics;

(2) At 2150 K, in the same atmosphere, the oxidation kinetics of ZrB_2_ + 20 v.% MoSi_2_ is linear due to the formation of gaseous SiO instead of silica; there is therefore no protective layer and the solid oxide layer on the substrate is mainly composed of porous monoclinic zirconia;

(3) The oxide layer formed at 1800 K is not only thinner but also harder and stiffer than the one formed at 2150 K. The glassy silica formed at 1800 K strengthens the oxide layer.

This observation demonstrates the utility of active cooling in maintaining ZBM thermal shields at a lower temperature (1800 K instead of 2150 K) in order to promote diffusion-controlled oxidation with a semi-parabolic regime: a silica layer would then form that would reduce the oxidation rate of the shield and simultaneously limit the fragility of the oxide layer. This would help preserve the structural integrity of the thermal shield over multiple exposures to the extreme thermo-chemical conditions of atmospheric re-entry, paving the way for the production of reusable thermal shields for spacecrafts. Our future work will focus on demonstrating the validity of such active cooling.

## Figures and Tables

**Figure 1 materials-18-01815-f001:**
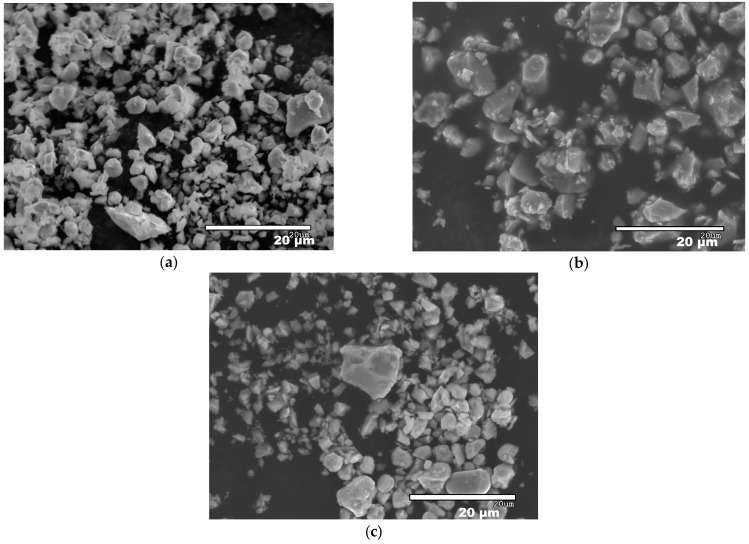
SEM images of as-received ZrB_2_ (**a**) and MoSi_2_ (**b**) powders and of the ball-milled mixture (**c**).

**Figure 2 materials-18-01815-f002:**
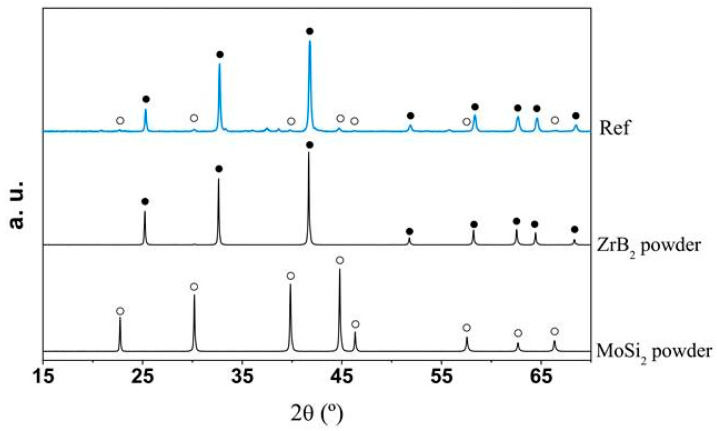
X-ray diffractograms of the as-received ZBM sample and the starting powders. 
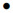
: ZrB_2_; 
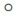
: MoSi_2_.

**Figure 3 materials-18-01815-f003:**
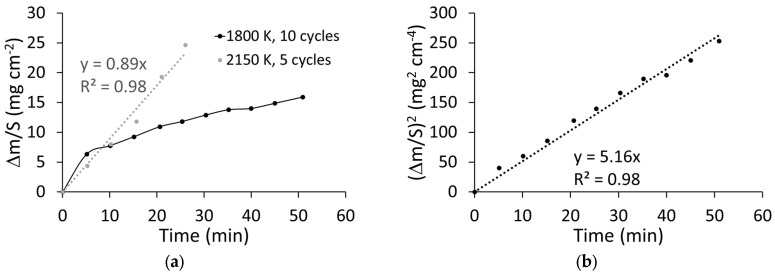
(**a**) Evolution of the weight change of the ZBM samples with time at 1800 and 2150 K. (**b**) Evolution of the square of the weight change of the ZBM samples with time at 1800 K.

**Figure 4 materials-18-01815-f004:**
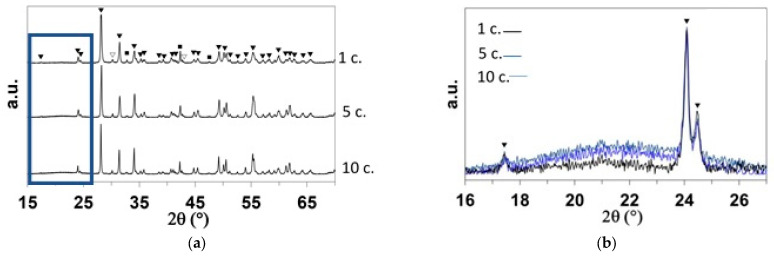
(**a**) X-ray diffractograms of one as-received ZBM sample (Ref.) and the ZBM samples after 1, 5, and 10 oxidation cycles at 1800 K in dissociated air at P = 1000 Pa. (**b**) Close-up of the 16–26° area (corresponding to the rectangular area in (**a**)). 
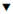
: m-ZrO_2_; 
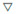
: t-ZrO_2_; 
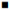
: MoB.

**Figure 5 materials-18-01815-f005:**
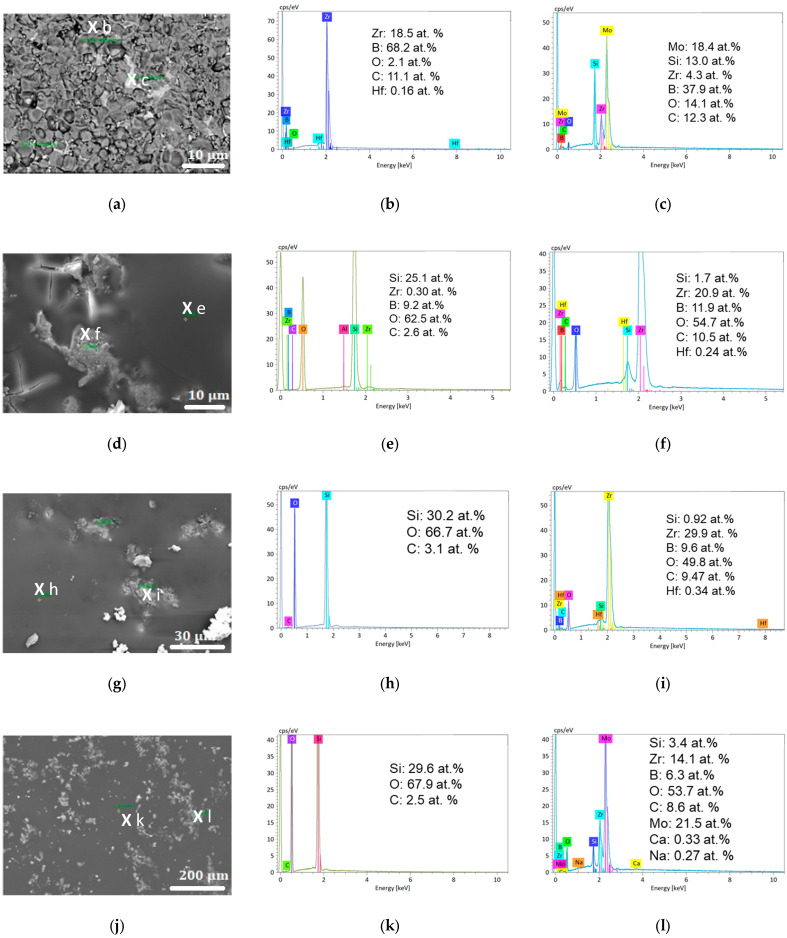
Surface SEM images (Secondary Electron) and the corresponding EDS, performed on the spots marked with an *X* in the images, of the ZBM samples as-received (**a**–**c**) and after one (**d**–**f**), five (**g**–**i**), and ten (**j**–**l**) oxidation cycles at 1800 K in dissociated air at P = 1000 Pa.

**Figure 7 materials-18-01815-f007:**
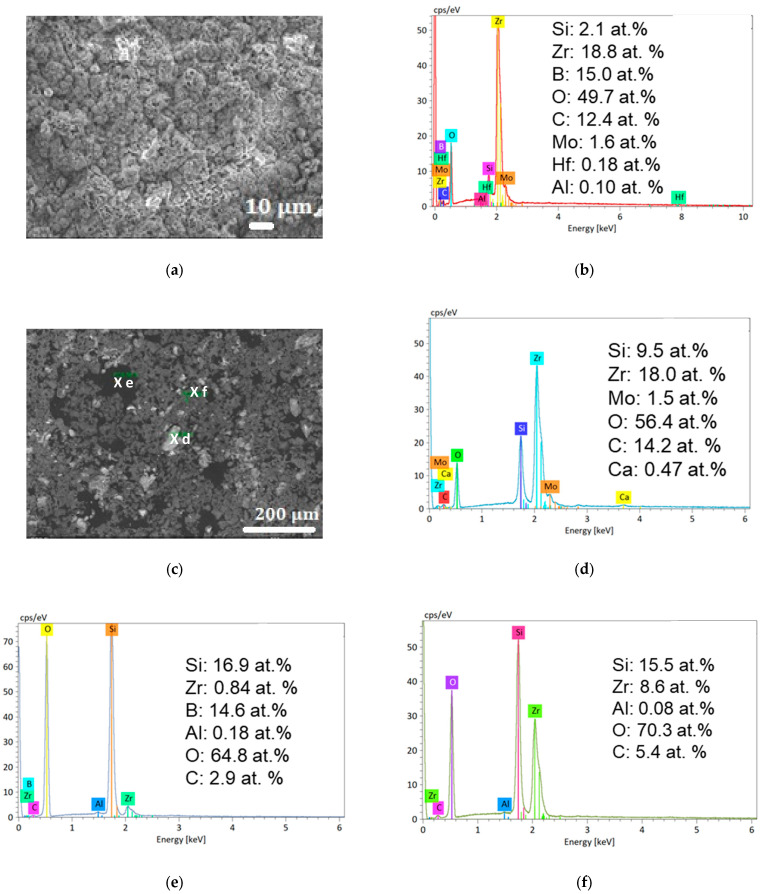
Surface SEM image (Secondary Electron) (**a**) and surface EDS (**b**) of a ZBM sample after one oxidation cycle at 2150 K in dissociated air at P = 1000 Pa. Surface SEM image (**c**) and EDS of the marked spots (**d**–**f**) of the ZBM sample after five oxidation cycles at 2150 K.

**Figure 8 materials-18-01815-f008:**
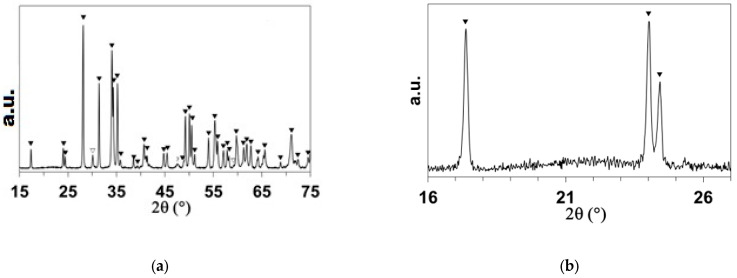
(**a**) X-ray diffractogram of the ZBM sample after 5 oxidation cycles at 2150 K in dissociated air at P = 1000 Pa. (**b**) Close-up of the 16–27° area. 
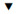
: m-ZrO_2_; 
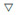
: t-ZrO_2_.

**Figure 10 materials-18-01815-f010:**
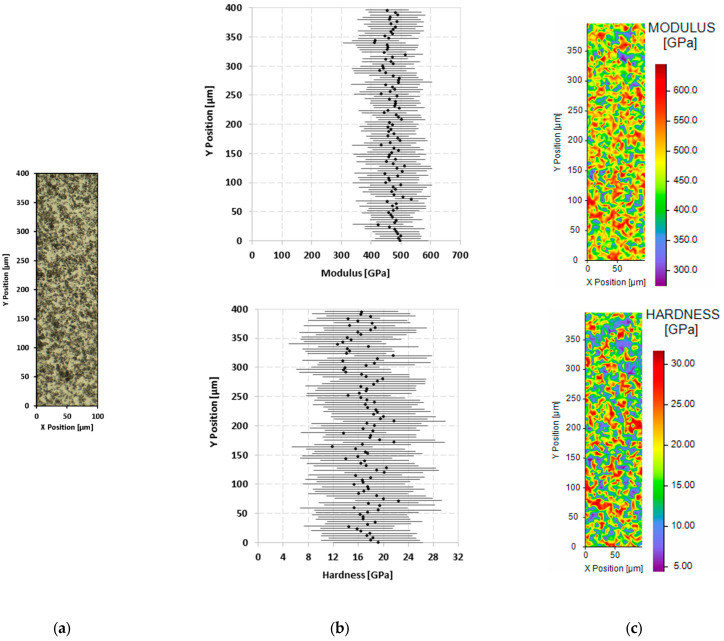
Results of nanoindentation test on the cross-section of the as-received ZBM sample: image from a numerical microscope (**a**), evolution of the averages and standard deviations of the elastic modulus and hardness along sample depth (Y axis) (**b**), and 2D mapping of the modulus and the hardness (**c**).

**Figure 11 materials-18-01815-f011:**
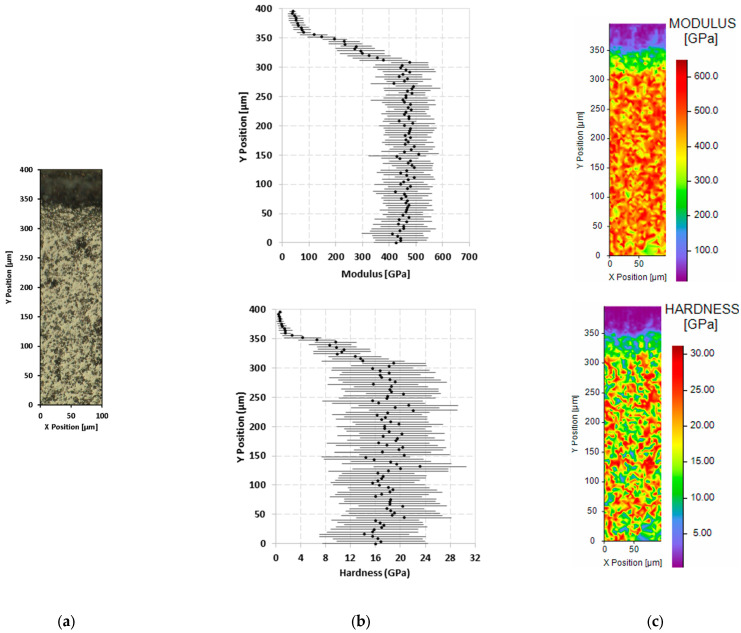
Results of nanoindentation test on the cross-section of the ZBM sample subjected to 10 oxidation cycles at 1800 K: image from a numerical microscope (**a**), evolution of the averages and standard deviations of the elastic modulus and hardness along the sample depth (Y axis) (**b**), and 2D mapping of the modulus and the hardness (**c**).

**Figure 12 materials-18-01815-f012:**
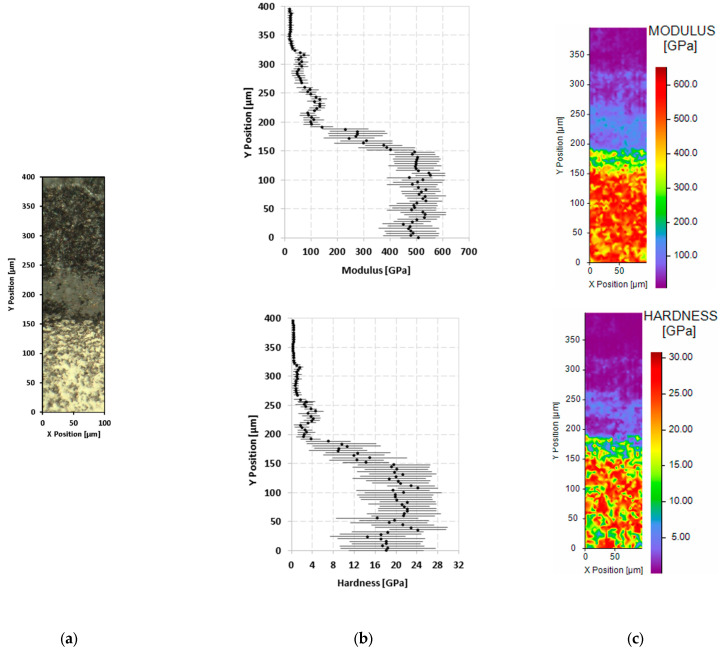
Results of nanoindentation test on the cross-section of the ZBM sample subjected to 5 oxidation cycles at 2150 K: image from a numerical microscope (**a**), evolution of the averages and standard deviations of the elastic modulus and hardness along the sample depth (Y axis) (**b**), and 2D mapping of the modulus and the hardness (**c**).

## Data Availability

The original contributions presented in this study are included in the article. Further inquiries can be directed to the corresponding author.

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
