# Peer review of "The Impact of Cyclic Oxidation in Dissociated Air on the Mechanical Properties of Freeze-Cast ZrB2/MoSi2 Ceramics"

_materials, 2025, doi:10.3390/ma18081815_

Round 1

Reviewer 1 Report

Comments and Suggestions for Authors

This manuscript reports work on ZrB2 ultrahigh temperature ceramics containing 20 vol.% MoSi2 and the cyclic oxidation of the material at 1800 K and 2150 K, where the material was exposed to free air, to investigate how it could withstand multiple re-entry tests. It is recommended that the material be used at lower than 1800 K to ensure its reusability. The reviewers suggest that this paper be published with minor modifications.

  1. Supplementary information on the specific suppliers and production batches of ZrB2 and MoSi2 powders is provided to ensure the repeatability of the experiment.
  2. The pattern of evolution of the weight change of the ZBM samples with time at 1800 and 2150 K is provided in Fig. 1a, but only the evolution of the square of the weight change of the ZBM samples with time at 1800 K is shown in Fig. 1b. It is recommended that the curve of the evolution of the square of the weight change of the ZBM samples with time at 2150 K be added to Fig. 1b.
  3. It is suggested that the presence of amorphous oxides is indicated at the bumps observed on Fig. 2b.
  4. In Fig. 3, the markings at the EDS point sweeps are too small.
  5. SEM and EDS plots after 5 and 10 cycles of oxidation at 2150 K are proposed to be added in Fig. 5.
  6. The physical and chemical mechanisms behind oxidation kinetics are discussed in depth, especially the fundamental reasons for different oxidation behaviors at 1800 K and 2150 K.
  7. In the conclusion part, the application prospect of this research achievement in reusable thermal protection system, as well as the possible research direction and technical challenges in the future are further discussed.
  8. The authors were advised to use references to supplement the introduction and the oxidation mechanism of ZrB2 ultrahigh temperature ceramics. Below ref proposed.

https://doi.org/10.1016/j.jeurceramsoc.2009.09.029
https://doi.org/10.1016/j.corsci.2021.109283

  1. The English writing of the manuscript should be polished.

Comments on the Quality of English Language

The English writing of the manuscript should be polished.

Author Response

This manuscript reports work on ZrB2 ultrahigh temperature ceramics containing 20 vol.% MoSi2 and the cyclic oxidation of the material at 1800 K and 2150 K, where the material was exposed to free air, to investigate how it could withstand multiple re-entry tests. It is recommended that the material be used at lower than 1800 K to ensure its reusability. The reviewers suggest that this paper be published with minor modifications.

1. Supplementary information on the specific suppliers and production batches of ZrB2 and MoSi2 powders is provided to ensure the repeatability of the experiment.

Our reply: (furtherly in italic) We thank the reviewer for this positive appreciation of our work. Supplier information was already provided in the original manuscript for both powders since they come from the same supplier, however, this has been made clearer now in the manuscript.

2. The pattern of evolution of the weight change of the ZBM samples with time at 1800 and 2150 K is provided in Fig. 1a, but only the evolution of the square of the weight change of the ZBM samples with time at 1800 K is shown in Fig. 1b. It is recommended that the curve of the evolution of the square of the weight change of the ZBM samples with time at 2150 K be added to Fig. 1b.

Fig. 1a presents the evolution of weight change with time for the two temperatures, which evidences the oxidation kinetics is linear at 2150 K (shown by a linear trend with a determination coefficient very close to 1: R2= 0.98) whereas it is not at 1800 K but looks semi-parabolic. Fig. 1b was included to support this hypothesis: at 1800 K, the square of the mass change evolves linearly with time, with another high determination coefficient (R2 = 0.98). We have made changes in the text to clarify why we used these two representations.

3. It is suggested that the presence of amorphous oxides is indicated at the bumps observed on Fig. 2b.

We have included a rectangular area in figure 2 a to indicate (as asked by another reviewer) the angular location of the bump data represented in figure 2 b. Obviously, XRD cannot identify the composition of the amorphous causing this large bump. However, as discussed in the manuscript, composition of the amorphous phases was given using EDS.

4. In Fig. 3, the markings at the EDS point sweeps are too small.

We have increased the size of the marks.

5. SEM and EDS plots after 5 and 10 cycles of oxidation at 2150 K are proposed to be added in Fig. 5.

As labeled in the legend, Fig. 5 (a-b) corresponds to the surface of the sample oxidized at 2150 K after one oxidation cycle, Fig. 5 (c-e) to the surface after 5 oxidation cycles. We have not performed more than 5 oxidation cycles at this temperature.

6. The physical and chemical mechanisms behind oxidation kinetics are discussed in depth, especially the fundamental reasons for different oxidation behaviors at 1800 K and 2150 K.

7. In the conclusion part, the application prospect of this research achievement in reusable thermal protection system, as well as the possible research direction and technical challenges in the future are further discussed.

We thank the reviewer for these very positive comments.

8. The authors were advised to use references to supplement the introduction and the oxidation mechanism of ZrB2 ultrahigh temperature ceramics. Below ref proposed.

https://doi.org/10.1016/j.jeurceramsoc.2009.09.029
https://doi.org/10.1016/j.corsci.2021.109283

We have added the corresponding references in the introduction.

9. The English writing of the manuscript should be polished.

We have paid for an English edition of the manuscript.

Reviewer 2 Report

Comments and Suggestions for Authors

Comments and suggestions for authors are given in the attached file.

Author Response

The main question addressed by the research presents an investigation by analyzing the damage evolution that would be induced by several cycles of oxidation below (at 1800 K) and above (at 2150 K) this temperature limit on freeze-casted ultra-high-temperature ceramics composed of ZrB2 with 20 vol.% MoSi2, ZBM samples. The investigations showed that at 1800 K, the formation of silica is expected to favor a diffusion-controlled oxidation regime, reducing the oxidation rate in time as the silica layer would act as a protective barrier while growing. At 2150 K, the absence of silica should favor a reaction-controlled oxidation regime with a constant oxidation rate.

This investigation aims at confirming there is a maximal working temperature below which ZBM could be reused in multiple operations paving the way to the production of reusable thermal shields for spacecrafts.

The text is clear and well-structured. Each section follows a logical sequence, facilitating the understanding of the concepts and results presented. The abstract is informative and accurately conveys the main conclusions.

The figures and tables are well-prepared and contribute significantly to the understanding of the results. The analyses presented through SEM, EDS, and XRD are clear and visually effective.

The experimental methods are described in detail, enabling reproducibility of the results.

Response (in italic afterwards): The authors thank the reviewer for the positive appreciation of our work

However, certain aspects, could be explored in greater depth:

Line 137, sentence: to explain a the influence of amorphous phase on oxidation mechanism and influence of of amorphous phase on... also how does content of crystalline The presence of an amorphous oxide is also evidenced in the bump observed on figure 2 (b), which increases with the oxidation time.

Comment: Content of amorphous phase and oxide phase on...to be introduced in order to explain a the influence of amorphous phase on oxidation mechanism and influence of of amorphous phase on oxidation mechanism.

We have used XRD in order to identify the crystalline compounds present before and after oxidation of our samples, and we noticed the presence of an amorphous phase. This is a qualitative approach, making an accurate quantification of phases is not necessary to evaluate the oxidation kinetics and its mechanisms since it is not the total amount of amorphous phase that plays a role in the oxidation kinetic, but he coverage provided by amorphous silica as it reduces the oxygen diffusion kinetic towards the substrate.

Line from 216-220, Sentence: figure 8 (b) these properties are homogeneously distributed along the depth (Y axis) with an average modulus and hardness of 471±89 GPa and 17±7.6 GPa, respectively. The important standard deviation of the measurements is mainly related to the local chemical inhomogeneity of the ZBM. Other effects are likely to contribute to this deviation such as the crystallographic orientation of the powders and the roughness of the analyzed surfaces.

Comment: Considering the crystallographic orientation of the powders Scans of 2q by these methods can be analyzed using the intensities of specific peak heights of interest normalized to those of the untextured sample. This method is widely favored and results in the Lodgering factor (LF or f(00l))

Again if this is so, Line from 227-228 Sentence: From a mechanical characterization method using the impulse excitation of resonant vibrations, R. J. Grohsmeyer et al. show that the modulus of ZrB2 progressively decreases with increasing MoSi2 content..

Comment: Authors must reconsider to determine the content of crystalline phases from xrd measurements to confirm the above statement.

We have not gone so far because the influence of the crystalline structure of the material on its mechanical properties was not the scope of our investigations. We have mainly focused on how the oxidation may affect these mechanical properties.

Line from 222-223, Sentence: for the sample oxidized at 2150 K, which can lead to inaccuracies in determining the contact surface between the Berkovich indenter and the sample..

Comment: How this can be overcome? If at all...But non the less..very large number of indentations per map provides satisfactory statistical measurements in agreement with those published.

One way to overcome this issue would be to introduce additional steps into the mechanical polishing procedure for the surfaces to be analyzed. These would consist of filling the pores with a polymer resin. This would allow us to clearly separate the mechanical properties of polished flat surfaces from those related to microporosity. This would perhaps also address the remark in line 177, by offering the possibility of analyzing the effect of pore size distribution on mechanical properties. The success is not guaranteed, but we could keep this method for another paper more detailed on this part.

It would be interesting to see how does thermal shock, thermal expansion coefficient, changes varying mentioned parameters, i.e. temperatures.

This kind of investigation would require equipment not available in our facilities. As an example, a recent investigation (https://onlinelibrary.wiley.com/doi/10.1155/2016/8346563) followed the evolution of anisotropic thermal expansion coefficient of ZrB2 using a synchrotron light source. Moreover, we can observe these authors have not performed thermal expansion coefficient measurement beyond 1150 K. Accessing such properties at high temperature is quite complicated!

Line-177, part of the sentence: abundant presence of microporosity: It would be also interesting to see how does the pore size distribution changes from oxide layer to non oxide layer – as important parameter for mechanical propertes.

Conclusion should be expanded to follow the main issues detected by this research.

We have expanded the conclusion as 3 reviewers on 4 claimed for this, adding a bulleted list of our main outcomes.

Compared to other published material these findings indicate the facts that at 1800 K, the formation of silica is expected to favor a diffusion-controlled oxidation regime, reducing the oxidation rate in time as the silica layer would act as a protective barrier while growing. At 2150 K, the absence of silica should favor a reaction-controlled oxidation regime with a constant oxidation rate. Conclusions are consistent with the evidence and arguments presented and they do address the main question posed. References in the introduction are not appropriate. No additional comments on the tables and figures.

All the references are appropriate and quality of the data is well provided.

No other issues were detected.

We thank the reviewer for these interesting comments.

Reviewer 3 Report

Comments and Suggestions for Authors

1) The introduction is very brief. Expand the introduction, with a focus in ZrB2/MoSi2 or similar ceramic systems, their microstructure, properties and oxidation resistance. At the end of the introduction try to explain the novelty of this work.

2) This work needs to focus more on the microstructural charecterization and analysis of the fabricated ZBM material. SEM images of the commercial powders, images of the powder after ball milling and images and elemental maps after SPS would help to understand the effect of the fabrication method on the microstructure of the ceramic material.

3) 5 and 10 oxidation cycles appear to be very short and not representing real conditions, especially since each cycle held the sample at the designated temperature only for 5 minutes. So this raises a question regarding the selected oxidation testing parameters.

4) How does the oxidation resistance compares to similar ceramic materials?

5) Please add a figure with higher magnification SEM images and elemental maps of the cross section of the material after oxidation, this would help to illustrate the oxidation effects and the formation of intermediate layers in the material.

6) Conclusions are very brief, please expand and try to make a reference to the main findings of this work.

My closing comment is that this is an interesting work that after a few improvements may be published.

Author Response

1) The introduction is very brief. Expand the introduction, with a focus in ZrB2/MoSi2 or similar ceramic systems, their microstructure, properties and oxidation resistance. At the end of the introduction try to explain the novelty of this work.

Response (in italic):This work is in the continuity on a previous article, reference 3, published in “Materials” and that can be found here (Open Access): https://doi.org/10.3390/ma17153818

Consequently we had to rephrase the introduction in order not to make plagiarism on the one of this previous article. The originality of the previous paper was to perform the oxidations in dissociated atmosphere, which is quite rare. The originality of this second paper is to go deeper while performing several oxidation cycles, as pointed in the sentence: “We hereby continue this investigation by analyzing the damage evolution that would be induced by several cycles of oxidation below (at 1800 K) and above (at 2150 K) this temperature limit on freeze-casted ZBM samples.”

2) This work needs to focus more on the microstructural charecterization and analysis of the fabricated ZBM material. SEM images of the commercial powders, images of the powder after ball milling and images and elemental maps after SPS would help to understand the effect of the fabrication method on the microstructure of the ceramic material.

Following your request, we have added SEM pictures of the powders before and after ball-milling. A DRX graph was added presenting the as-produced samples and comparing its crystallography to the ones of the initial powders. Complementary characterizations can be found in the reference 6 (https://doi.org/10.1016/j.jeurceramsoc.2024.116966), as pointed in the new added sentences.

3) 5 and 10 oxidation cycles appear to be very short and not representing real conditions, especially since each cycle held the sample at the designated temperature only for 5 minutes. So this raises a question regarding the selected oxidation testing parameters.

We have added some explanations on the reasons why we have chosen such an experimental protocol.

The period of time when a spacecraft is exposed to an oxygen plasma while entering the atmosphere is really short indeed. The time period is the one we have chosen in our former work https://doi.org/10.3390/ma17153818 You can nevertheless observe as we did that the changes induced by the oxidation are sufficient enough to be visible even after one oxidation cycle.

We have performed 5 oxidation cycles at 2150 K as this is satisfactory enough to show no protective oxidation occurs. We have performed 10 oxidation cycles at 1800 K (corresponding to 10 reentries with the same thermal shield, which is important as currently the shields are replaced after each flight) to demonstrate how a thermal shield could support passive oxidation at this temperature.

4) How does the oxidation resistance compares to similar ceramic materials?

In our previous paper https://doi.org/10.3390/ma17153818, we had investigated how the oxidation behavior of ZrB2/MoSi2 could compare to other ceramic material. As written in the discussion of this paper: “Regarding previous investigations reported in the scientific literature, there are very few examples of the same compounds being oxidized under similar conditions. In a recent article [https://doi.org/10.31224/osf.io/n9qd4 ], Saha et al. investigated the cyclic oxidation of ZrB2 + 20 vol.% MoSi2 for 6 h in air at temperatures from 1373 to 1673 K and identified the presence of ZrO2 and SiO2, together with molybdenum oxide. Cyclic oxidation of the samples followed linear oxidation kinetics from 1373 to 1623 K, while at 1673 K, the oxidation kinetics were parabolic due to the protective action of SiO2. These authors expressed interest in continuing these investigations at higher temperatures, as undertaken here. We can confirm that silica forms at up to 1800 K but disappears at higher temperatures due to active oxidation.”.

Indeed the originality of the oxidation test and of the method used for the elaboration of the sample make it difficult to find relevant comparison in the literature.

5) Please add a figure with higher magnification SEM images and elemental maps of the cross section of the material after oxidation, this would help to illustrate the oxidation effects and the formation of intermediate layers in the material.

We do not understand here to which SEM images you are referring to. The magnification of the cross-section was chosen to evidence any layer that have formed, and regarding surface SEM imaging, we have chosen a magnification that evidenced the different phases to analyze.

6) Conclusions are very brief, please expand and try to make a reference to the main findings of this work.

We have expanded the conclusion as suggested by several reviewers. A bulleted list resumes our main results.

My closing comment is that this is an interesting work that after a few improvements may be published.

We thank the reviewer for this positive appreciation of our work.

Reviewer 4 Report

Comments and Suggestions for Authors

   The paper ” The Impact of the Cyclic Oxidation in Dissociated Air on the Mechanical Properties of Freeze-Casted ZrB2/MoSi2 Ceramics’’ can be published in Materials after some corrections:

  1. The abstract presents the information somewhat chaotically. The idea of ​​this study is somehow understood but it is not presented in the best form.
  2. The Introduction chapter presents too few aspects and confirms that the authors have not gone through the specialized literature with interest to explain exactly what the current state of research is and what exactly this work adds. In other words, what is the scientific progress brought by this work, if it is not known exactly what the latest and most relevant achievements in this field are. I will send you some works to give you an idea of ​​how comprehensive an introduction chapter should be (DOI:10.3390/cryst13020245 ; doi.org/10.1016/j.matpr.2022.02.254).
  3. The Introduction contains only 3 bibliographical references, so consulting such a small number of materials/articles/etc. no concrete conclusions can be drawn about the technological level in this field.
  4. In the Materials and Methods chapter, it would be advisable that, for a better understanding, the chemical compositions be put in a table.
  5. It would be useful to add at least one picture from the time of the analyses in which the samples used can be seen. It should also be mentioned in detail regarding the preparation of the experimental samples.
  6. In figure 2, a legend should be added in which the compounds and the symbol used for them should be mentioned.
  7. For figure 2, (b), it should be redone, the displayed graph does not have a good quality as it is not very well defined
  8. At least, figures 3,4,5,8,9,10 are extremely large, the figures and the text should have some proportionality in the paper. It seems that there are more figures than explanations.
  9. The images taken in cross-section should be made, additionally, at higher magnifications in order to be able to see exactly the formed connection.
  10. The Conclusions Chapter is not properly done. It should present point by point, the main concrete and remarkable conclusions from this work. At least 5 conclusions that clearly expose the results of this work.
  11. The bibliographical references are found in a relatively small number, which leads to the idea that the current state of research in the field is not very well mastered.
  12. Overall, it seems that no relevant effort was made for the work and it presents numerous errors.
  13. I believe that this work should be rewritten with great care to be at the highest possible level of quality. There are useful analyses and tests but they are not exposed exactly ok.

Comments on the Quality of English Language

I am not qualified to assess the quality of the English language.

Round 2

Reviewer 4 Report

Comments and Suggestions for Authors As far as can be seen, the author has taken into account most of the recommendations made so that the process of evaluating and publishing the work can continue. However, a check should be made of compliance with the template and, at the same time, a rearrangement of the figures in terms of their dimensions.